green chemistry/materials science

waste PET, glycolysis, waterborne alkyd, waterborne alkyd-amino baking coatings

**Authors for correspondence:**
Ding Yong-bo
e-mail: yongboding@jxstnu.com.cn
Shen Liang
e-mail: liangshen@jxstnu.com.cn

This article has been edited by the Royal Society of Chemistry, including the commissioning, peer review process and editorial aspects up to the point of acceptance.

# Preparation and characterization of waterborne alkyd-amino baking coatings based on waste polyethylene terephthalate

Xia Yun, Yang Xin-yi, Gong Dun-hong, Ding Yong-bo and Shen Liang

The Department of Coatings and Polymeric Materials, College of Chemistry and Chemical Engineering, Jiangxi Science and Technology Normal University, Nanchang 330013, China

DY, 0000-0002-9203-0412

The recycling of polyethylene terephthalate (PET) is the most attractive method for PET waste management because it not only decreases the load on landfill space, but also provides opportunities for reducing the use of raw petrochemical products. Therefore, in this investigation, neopentyl glycol is used for alcoholysis of waste PET, and glycolyzed PET product was applied for preparation of the waterborne alkyd resin. Furthermore, the waterborne alkyd-amino baking coatings were prepared from the waterborne alkyd based on glycolyzed waste PET and melamine formaldehyde resin and applied on tinplate. The waterborne alkyd-amino resin films showed excellent adhesion, balanced hardness and flexibility, high gloss and outstanding chemical resistance except for alkali resistance owing to hydrolysis of ester bonds.

## 1. Introduction

Waste polyethylene terephthalate (PET) bottlers create a large amount of garbage in our daily life, because waste PET is not easily decomposed in nature, and huge quantities of waste PET have led to dangerous environmental contamination [1]. Recycling of PET appears to be the most attractive method for the associated waste management because the recycling of waste PET bottles not only solves the problem of solid-waste, but also its depolymerized products can be used as raw materials for

some industrial products [2]. Existing PET recycling methods include: (i) landfill disposal, (ii) mechanical recycling, (iii) chemical recycling, and (iv) incineration [3]. According to the standards of sustainable development, among the above-mentioned four kinds of recycling methods, the only acceptable recycling method is the chemical method because the raw materials it produces can be used to synthesize other polymers [3,4].

PET containing ester groups can be hydrolysed [5], alcoholized [6], glycolyzed [7–12] and ammonolysed [13–16] in the presence of water, alcohol, glycols and amines. Glycolysis is an important and commercial process for the chemical depolymerization of PET. The main products are terephthalic acid (TPA) bis(2-hydroxyethyl) ester and other oligomers obtained from glycolysis of waste PET bottles with ethylene glycol (EG) [17] or neopentyl glycol (NPG) [18]. These products (monomers) can be used to prepare polyester [8,11–12,19,20], alkyd [10,14,21–24], polyurethane (PU) [18], epoxy [9] or adsorbent for basic dyes [25], etc., and possess identical properties to those prepared from its equivalent monomers.

Alkyd resin is characterized with excellent adhesion, high gloss, good flexibility, outstanding wettability, cheapness and simple application, which has been used for surface coating for more than 80 years [26]. Alkyd resin has been the workhorse of the binders for surface coatings. Its advantage is that it does not depend on petroleum resources [27]. Besides, alkyds tend to provide coatings with high gloss and good adhesion [26]. But traditional alkyds are mainly used in solvent-borne coatings. The volatile organic component (VOC) emission is large, so reducing VOC in coatings is a challenge for coatings chemists and technicians. When it comes to environmental protection, it is vital to develop waterborne alkyd resins [28–34].

Up to now, waste PET has been used to synthesize alkyd resin, but it is rare to synthesize waterborne alkyd resin. In [23], waste PET has been used to prepare waterborne alkyd resin, however, surfactants are added in order to improve its water solubility. Therefore, in this paper, glycolyzed PET, and high-quality, low-cost tall oil fatty acid (a by-product of kraft pulping) were used as raw materials to prepare low-cost and environmentally-friendly bio-based waterborne alkyd resins by introducing water-based monomer trimellitic anhydride (TMA). Furthermore, waterborne alkyd-amino resin films were also prepared with waterborne alkyd resin based on waste PET and amino resin.

# 2. Experimental method

## 2.1. Materials

All raw materials applied can be purchased from the market, except for the waste PET which was acquired form discarded soft drink bottles. The discarded PET bottles were first washed with water and dried, then they were cut into smithereens of about 10 mm². It was then washed with acetone and dried at 100°C for 8 h. Tall oil fatty acids (TOFA) were supplied by Anhui refined oil and fat Co., Ltd. NPG, isophthalic acid (IPA), benzoic acid (BA), TMA, zinc acetate was obtained from Aladdin. Pentaerythritol (PE), xylene, triethanolamine, N, N-Dimethylethanolamine (DMEA), dipropylene glycol butyl ether (DPNB) and butyl cellosolve (BCS) were acquired from Jiangxi Pinghai Biotechnology Co., Ltd. Titanium dioxide (rutile type) was purchased from Du Pont. Partial etherified amino resins (325) and commercial alkyd were purchased from TOD Resins. The dispersing agent BYK-190, anti-settling agent BYK-420, anti-foaming agent BYK-024 and levelling agent BYK-381 were supplied by BYK additives & instruments.

## 2.2. Glycolysis of waste PET

The glycolysis of waste PET bottles was accomplished according to reference [18], and the glycolysis equipment used in this experiment is also the same as that used in reference [18]. In this experiment, 156 g of NPG, equivalent to 1.50 mol, 48 g PET, equivalent to 0.31 mol repetitive unit in the PET molecular chain, 1.02 g zinc acetate, equivalent to 0.005 mol were added to the four-necked flask. The reaction mixture was heated at 220°C for 6 h under the protection of nitrogen. As the temperature increased, the PET chips began to melt, eventually resulting in a homogeneous milky white liquid. After discharging, it was cooled to room temperature and turned into a white opaque solid.

**Table 1.** Formulation of alkyd resin.

| material | weight (g) | $e_a$ | $e_b$ | $m_0$ | step |
|---|---|---|---|---|---|
| TOFA | 89.99 | 0.3191 | | 0.3191 | step 1 |
| PE | 24.94 | | 0.7328 | 0.1832 | |
| glycolyzed PET | 49.88 | | 0.7298 | 0.3649 | |
| IPA | 47.68 | 0.574 | | 0.287 | |
| BA | 3.5 | 0.0287 | | 0.0287 | |
| TMA | 20 | 0.3123 | | 0.1041 | step 2 |
| total | 235.99 | 1.2341 | 1.4626 | 1.2870 | |

## 2.3. Synthesis of medium oil waterborne alkyd resin

Medium oil waterborne alkyd resin (43%) was prepared from TOFA, glycolyzed PET, phthalic anhydride (PA), PE, IPA, BA, and TMA. The K constant was around 1.05, the $R$ value (ratio of total -OH groups to total -COOH groups) was about 1.19. The feed ingredients were listed in table 1.

In step 1, the ingredients were added to a four-necked glass flask equipped with a mechanical stirrer, thermometer, water/oil separator and a rubber stopper. Xylol was applied as the azeotropic solvent. The mixture was slowly heated, the temperature of the reaction was kept constant at 230–235°C in the end until the acid number (AN) was less than 10 mg KOH $g^{-1}$. The reaction was followed with AN which was measured through titration of samples dissolved in methylbenzeneethyl alcohol solution with 0.1M potassium hydroxide solution.

In step 2, the reaction temperature was reduced to 185°C, and the added TMA then reacted, staying at 185°C until the AN was 40–60 mg KOH $g^{-1}$.

Finally, the prepared alkyd resin was cooled to 150°C and diluted to 80% solid content by adding DPNB as a co-solvent, then triethanolamine was used as a neutralizing agent to neutralize carboxylic acid functional groups. The amount of an amine (mass: $m_{amine}$; molar mass: $M_{amine}$) for complete neutralization of an alkyd resin (mass: $m_{alkyd}$) can be calculated from the acid number (AN, mg KOH $g^{-1}$) of the alkyd [35]:

$$m_{amine} = \frac{M_{amine} \cdot AN \cdot m_{alkyd}}{56100}.$$

## 2.4. Preparation of waterborne alkyd-amino baking coating

The pigment pastes were grounded through a sand mill, mixing 70 wt% rutile titanium dioxide, 21 wt% deionized water, 8 wt% BYK-190, 0.6 wt% BYK-024, 0.4 wt% BYK-381 until the TiO$_2$ fineness grade reached less than 20 μm.

The alkyd-amino baking coating formulation is listed in table 2. Firstly, the alkyd resin based on waste PET (for dilution, the desired amount of water was added while stirring vigorously at room temperature; the final content of solids in the water reducible alkyd resin was 50% by weight) and 325 amino resin were premixed for 5 min; secondly BCS was added to reduce the viscosity of the resins, and DMEA was used to adjust pH to a value of about 7, then deionized water was applied to adjust the viscosity of the system to the appropriate position. BYK-024 and BYK-381 were added, and finally pigment paste was added. All of them were dispersed for about 30 min at 500–600 rpm. The alkyd-amino resin film was prepared by scraping the coating with a 100 μm wire rod on a polished tinplate. It was then heated at 150°C for 0.5 h in an oven and its properties were determined. Meanwhile, commercial alkyd was also used to prepare alkyd-amino resin film in the same way.

## 2.5. Characterization

### 2.5.1. $^1$HNMR analysis of waterborne alkyd

The $^1$HNMR spectral measurements of the waterborne alkyd were carried out on a Bruker AVANCE IIIHD (400 MHz). The spectra were recorded with deuterated chloroform at room temperature. The chemical shifts were expressed in ppm values according to those of tetramethyl silane.

**Table 2.** Formulation of alkyd-amino baking coating.

| raw material | weight (g) |
| --- | --- |
| alkyd resin (50% solid content) | 40 |
| 325 amino resin (80% solid content) | 6.25 |
| BCS | 4 |
| DMEA | 0.6 |
| deionized water | 20.3 |
| anti-foaming agent BYK-024 | 0.2 |
| levelling agent BYK-381 | 0.25 |
| pigment paste ($TiO_2$, 70%) | 28.4 |
| total | 100 |

### 2.5.2. GPC analysis of waterborne alkyd

The molecular weight of the waterborne alkyd was examined by a Waters 1515 gel permeation chromatograph (GPC). Tetrahydrofuran was applied as a mobile phase at $1.0 \, ml \, min^{-1}$ of flow rate at 25°C. The molecular weights were calculated with reference to the calibration curve of standard polystyrene.

### 2.5.3. Rheological measurements of waterborne alkyd

For the sake of determining the rheology of the waterborne alkyd, a DHR-2 rheometer from TA Instruments Ltd. was used. The dynamic rheological properties were determined at shear rate varying from $6.60 \, s^{-1}$ to $1995.12 \, s^{-1}$. All measurements were carried out at 25°C.

### 2.5.4. Particle size analysis of waterborne alkyd

In order to determine the particle size of the waterborne alkyd, a Nicomp 380DLS from Malvern Instruments was used at room temperature. For this, a solution of waterborne alkyd (1 wt%) was used.

### 2.5.5. Testing of waterborne alkyd-amino resin films

Hardness was measured according to ASTM D3363 using a pencil hardness tester (BEVS1301 pencil hardness tester from BEVS industrial Co., Ltd.), the adhesion of the prepared waterborne alkyd-amino baking coatings to the substrate were characterized according to ASTM D 3359 using the crosshatch adhesion method (BEVS 2202 cross hatch cutter from BEVS industrial Co., Ltd.), the flexibility was measured using a cupping tester (BEVS automatic cupping tester from BEVS industrial Co., Ltd.) (ISO 1520), the flexibility was also measured according to ASTM D 522 using a cylindrical mandrel bend tester (BEVS1603 cylindrical mandrel bend tester from BEVS industrial Co., Ltd.), the impact resistance was measured according to ASTM D 2794 using an impact tester (BEVS impact tester from BEVS industrial Co., Ltd.), and gloss was measured at an angle of 60° (ASTM D 2457) using a gloss meter (BEVS 60° Glossmeter from BEVS industrial Co., Ltd.).

The chemical resistance of waterborne alkyd-amino baking coating was studied as below. A piece of absorbent cotton swollen in a chemical medium was put on a film-coated tin plate, which was covered with a culture dish and kept in a laboratory with constant temperature and humidity (23 ± 2°C, 50% ± 5% RH) for 24 h [36].

## 3. Results and discussion

### 3.1. Characterizations of waterborne alkyd

The chemical structure of prepared waterborne alkyd is shown in figure 1a, and it was also characterized by $^1$HNMR (figures 1b). Results of $^1$HNMR analysis of waterborne alkyd are as below:

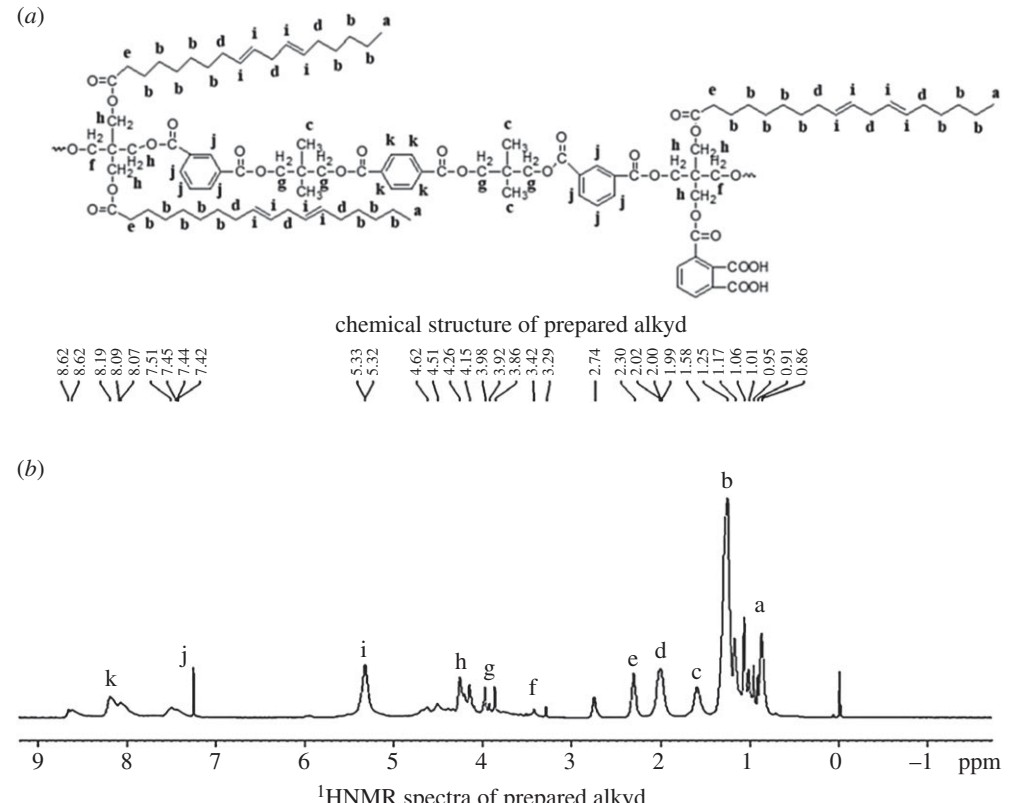

Figure 1. Chemical structure (*a*) and ¹HNMR spectra (*b*) of prepared alkyd.

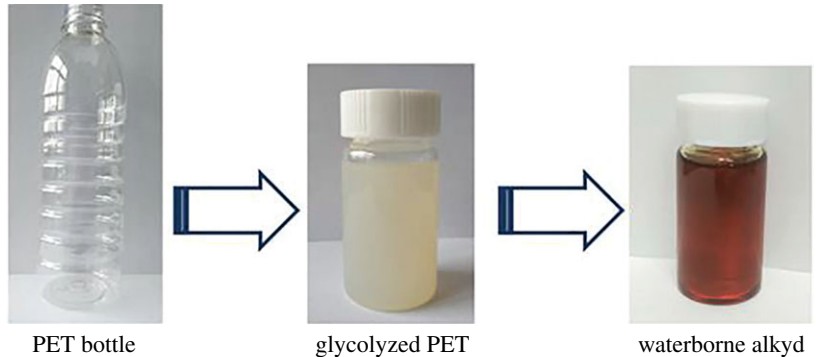

Figure 2. The appearances of the glycolyzed PET and waterborne alkyd.

$\delta$ 0.86–1.17 (a, -C$\underline{H}_3$, TOFA moiety), $\delta$ 1.25 (b, -C$\underline{H}_2$-, TOFA moiety), $\delta$ 1.58 (c, -C$\underline{H}_3$, NPG moiety), $\delta$ 1.99–2.02 (d, -C$\underline{H}_2$-CH=CH-, TOFA moiety), $\delta$ 2.30 (e, -C$\underline{H}_2$-COO-, TOFA moiety), $\delta$ 3.42 (f, -C$\underline{H}_2$-OH, PE moiety), $\delta$ 3.86–3.98 (g, -C$\underline{H}_2$-OH, NPG moiety), $\delta$ 4.15–4.62 (h, -COO-C$\underline{H}_2$-, PE moiety), $\delta$ 5.32–5.33 (i, -C$\underline{H}$=C$\underline{H}$-, TOFA moiety), $\delta$ 7.42–7.51 (j, -C$_6\underline{H}_4$-, IPA moiety), $\delta$ 8.09–8.19 (k, -C$_6\underline{H}_4$-, TPA moiety).

The appearances of the glycolyzed PET and waterborne alkyd are shown in figure 2. The prepared glycolyzed PET was a light yellow viscous substance at high temperature and white solid substance under normal temperature. The colour of the synthesized alkyd resin was yellowish-brown, thanks in part to the brownish colour of the glycolyzed PET applied as reactant. Moreover, the severity of high temperature transesterification/esterification led to the yellowish-brown of the prepared waterborne alkyd [37].

The size distribution of resin dispersions can explain the reaction process of resin polymerization to a certain extent, and is an important index to measure the properties of polymers. Therefore, it is necessary to determine and analyse the size and distribution of resin dispersions. The droplet size distributions of the prepared waterborne alkyd are indicated in figure 3. From figure 3, we can see that the average particle size of the prepared waterborne alkyd resin is 72.3 nm, which is in the range of particle size of colloidal solution (1–100 nm), indicating that the water dispersibility of the prepared waterborne alkyd is excellent [38].

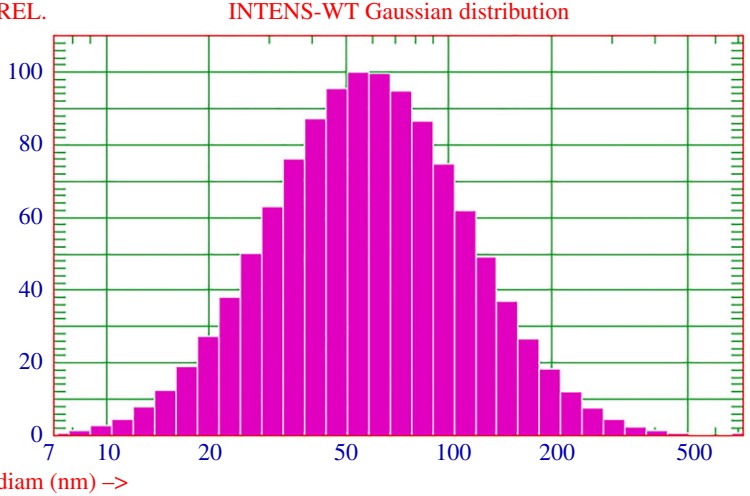

**Figure 3.** The droplet size distributions of the prepared waterborne alkyd.

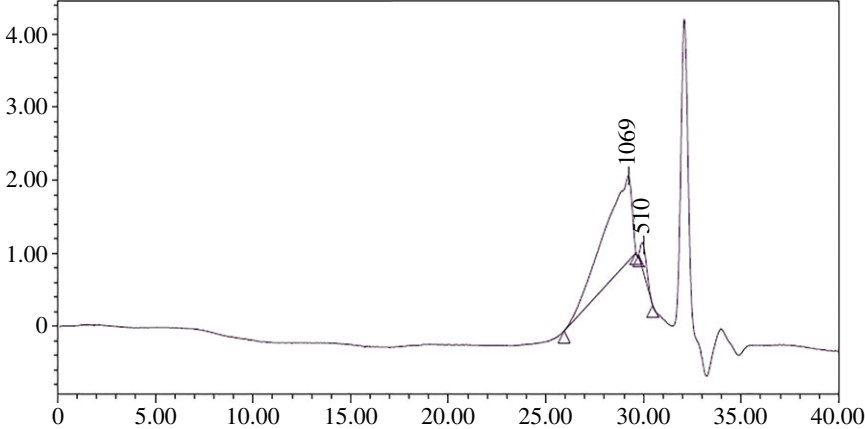

**Figure 4.** GPC chromatograms of the synthesized alkyd resin.

The average molecular weight and its distribution of polymers are very important for the properties of polymers. The average molecular weight and its distribution of the prepared waterborne alkyd resins were identified through gel permeation chromatography. The results are displayed in figure 4. The calculated values of $M_n$, $M_w$ and the polydispersity index (PDI) are 1849 g mol$^{-1}$, 2626 g mol$^{-1}$ and 1.42, respectively. It is common knowledge that alkyd resins with narrow molecular weight distribution exhibit better properties [39].

Therefore, we speculate that the waterborne amino baking varnish prepared by the waterborne alkyd resin will have excellent mechanical properties.

Thixotropic property is vital for effective processing to be achieved. Therefore, rheological behaviour of the prepared waterborne alkyd resin has been studied (figure 5). As is shown in figure 5, the viscosity of waterborne alkyd resin decreases with the increase of shear rate. This is because waterborne alkyd resin droplet network is relatively stable at a small shear rate, and the shear force cannot open the droplet network of waterborne alkyd resin, resulting in a higher viscosity. However, when the shear rate increases gradually, the droplet network of waterborne alkyd resin is gradually destroyed by the shear force and the viscosity decreases, which shows thixotropy of shear thinning [40].

## 3.2. Characterizations of waterborne alkyd-amino resin films

In order to achieve the long-term protection function of the coatings, various properties have to be observed. An excellent adhesion of organic coatings to the substrate is the precondition for the protection of coatings. The crosshatch adhesion method was used to measure the adhesion of the waterborne alkyd-amino baking coatings to the tinplate. Strength of adhesion is graded by using

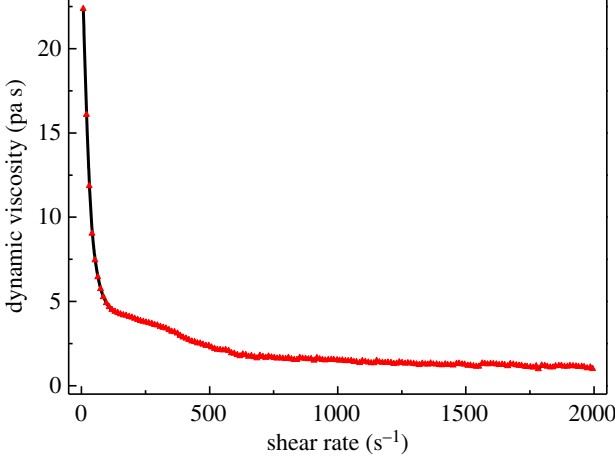

**Figure 5.** Dynamic viscosity versus shear rate of the prepared alkyd resin at a temperature of 295 K.

**Table 3.** The physical properties of alkyd-amino resin films and comparative literature.

| properties | value | | |
|---|---|---|---|
| | in this work | commercial alkyd resin | comparative literature [23] |
| film thickness, μm | 47.6 | 35 | — |
| hardness | 2 H | 3 H | 105 s |
| flexibility | 2 mm | 3 mm | — |
| adhesion | 5 | 5 | — |
| impact resistance | 50 cm kg | 50 cm kg | — |
| gloss, 60° | 92.3 | 78.2 | — |

this method. The reproducible adhesion test results (table 3) demonstrated that waterborne alkyd-amino baking coatings had an excellent adhesion to tinplate owing to the fact that the alkyd-amino baking coatings edges were completely smooth, with no partial delamination. The excellent adhesion could stem from the reaction of residual hydroxyl functional groups in the molecular structure of waterborne alkyd resin with tinplate and homogeneous cross-linking through alkyd-amino baking coatings [18].

The coatings will undergo many different deformations during their service life, which must be withstood without loss of adhesion, physical damage or performance impairment. In the broadest sense, material with excellent flexibility can meet most of the mechanical deformations [41]. The mandrel bending test (table 3) with a cylindrical mandrel showed that the cross-linked films were relatively flexible due to the diameter at which the waterborne alkyd-amino baking coating cracks after stress is less than 2 mm. Apart from one-dimensional elongation until cracking, two-dimensional stretching (such as cupping test) are also applied to determine the flexibility of alkyd-amino baking coatings. After the cupping test, the alkyd amino drying coating has no cracking and delamination although the tinplate has cracked and the distance covered by the plunger is 10 mm (figure 6). The results clearly show that the alkyd-amino baking coatings were relatively flexible and adherent in the thickness of the coatings used [42].

The ball impact tester was also applied to measure the dynamic deformation under impact stress. As shown in table 3, the impact resistance is more than 50 cm kg, and the results showed that the impact resistance of alkyd-amino baking coatings is excellent.

Coatings with excellent properties must not only possess outstanding adhesion and fine elasticity, but also possess the required hardness. The hardness of the waterborne alkyd-amino baking coatings was 2H, which meets the general requirements of hardness application. Therefore, waterborne alkyd-amino baking coatings offer a good balance of hardness and flexibility.

The gloss characteristics are very necessary when the aesthetic or decorative appearance of coatings is of great significance. As presented in table 3, the gloss determined at 60° was found to be superior, which can be attributed to uniform cross-linking of the whole film and favourable compatibility of reactants [43], which lead to fine film forming and levelling, since a perfectly smooth surface gives high gloss.

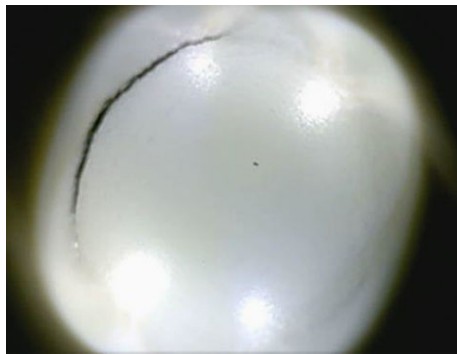

**Figure 6.** Cupping test of the alkyd-amino baking coatings.

**Table 4.** Chemical resistance properties of the waterborne alkyd-amino resin films and comparative literature.

| chemical medium | influence | | |
| --- | --- | --- | --- |
| | in this work | commercial alkyd resin | comparative literature [23] |
| water | completely unaffected | completely unaffected | no change |
| NaCl (2%) | completely unaffected | completely unaffected | no change |
| $H_2SO_4$ (2%) | completely unaffected | completely unaffected | no change |
| NaOH (2%) | film faintly swelled | severe blistering | destroyed |
| xylene | completely unaffected | completely unaffected | no change |
| acetone | completely unaffected | completely unaffected | no change |

Meanwhile, both of the physical properties of amino baking coating prepared with commercial alkyd resin and another kind of alkyd resin in the reference [23] are also listed in table 3. It can be concluded that the physical properties of the film prepared with our alkyd resin is comparable to the two above alkyd resins.

## 3.3. Chemical resistance of waterborne alkyd-amino resin films

As shown in table 4, the waterborne alkyd-amino resin films were unaffected by $H_2O$ and 2% NaCl solution. The resin film is very important for the water and salt solution resistance in order to ensure the durability of the resin film in use [44]. Generally, alkyd resin films are seldom affected by acid solutions. When sodium hydroxide solution was used to test the resistance of waterborne alkyd amino resin films, swelling and whitening of the paint film were found. Poor alkali resistance of waterborne alkyd amino resin films may be due to hydrolysis of ester bonds in its molecular structure under alkaline conditions [45]. Xylene is used to test whether there are solvents in waterborne alkyd amino resin films. The excellent xylene resistance results show that the dry film of the waterborne alkyd amino resin films contains no solvents. Polar solvents like acetone are often applied to evaluate the extent of curing and cross-linking of coatings. The excellent acetone resistance results show that there is a high degree of cross-linking in waterborne alkyd amino resin films between the prepared waterborne alkyd based on waste PET and amino resin [46].

Similarly, the film resistance of alkyd-amino baking coating prepared with commercial alkyd resin and another kind of alkyd resin in the reference [23] were also listed in table 4. And it can be seen that alkyd-amino resin film in this work has better alkali resistance, and the remaining resistance properties (water, acid, salt solution and organic solvents) are also comparable to those of the two above alkyd resins.

## 4. Conclusion

The waste PET was glycolyzed by NPG with zinc acetate as a catalyst. Glycolyzed PET and tall oil fatty acid were used to prepare low-cost and environmentally-friendly bio-based waterborne alkyd resins.

Furthermore, waterborne alkyd-amino baking coating was also prepared with waterborne alkyd resin based on waste PET and amino resin. The film properties of waterborne alkyd-amino baking film were investigated. Meanwhile, experimental results were compared with similar studies and commercial alkyd. The following conclusions can be drawn from the obtained results.

The physical properties (hardness, flexibility, adhesion, impact resistance and gloss) of waterborne alkyd-amino baking film in this work are comparable to those of amino baking coating prepared with commercial alkyd resin and another kind of alkyd resin in the comparative literature.

Three kinds of waterborne alkyd film are resistant to water, acid and salt solution until 24 h, and the alkyd film in this work has better alkali resistance.

The alkyd-amino baking film in this work showed good resistance to organic solvents (acetone and xylene).

Therefore, the use of waste PET in water-soluble coatings systems not only reduces the cost of coatings, but also opens up a new market for recycled PET, which may contribute a promising method for management of waste PET. So, it is an interesting method for environmentally friendly and efficient disposal of waste PET.

Data accessibility. Our data are deposited in the Dryad Digital Repository at: https://doi.org/10.5061/dryad.c7rg5tt.
Authors' contributions. D.Y.-B. and S.L. designed the study. X.Y., Y.X.-Y. and G.D.-H. prepared all samples for analysis. X.Y. and D.Y.-B. collected and analysed the data. X.Y., D.Y.-B. and S.L. interpreted the results and wrote the manuscript. All authors gave final approval for publications.
Competing interests. We declare we have no competing interests.
Funding. Financial support came from the National Natural Science Foundation of China (grant no. 51563011) and scientific research foundation of Jiangxi Science and Technology Normal University (grant no. 3000990309).
Acknowledgements. We thank Zhong J., for his assistance with thixotropic property analyses.

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
