## [Reviewer comments · Royal Society Open Science]

Review History

RSOS-191447.R0 (Original submission)

Review form: Reviewer 1

Is the manuscript scientifically sound in its present form?

Yes

Are the interpretations and conclusions justified by the results?

Yes

Is the language acceptable?

No

Do you have any ethical concerns with this paper?

No

Have you any concerns about statistical analyses in this paper?

No

Recommendation?

Accept with minor revision (please list in comments)

Comments to the Author(s)

In this paper, the authors partially used the glycolyzed PET product to synthesize waterborne alkyd resin. The resin was further fabricated as waterborne alkyd-amino baking coatings with better mechanical properties. Although the glycolyzed PET was ever used to synthesize alkyd, it is seemed that this paper is the first time to employ the recycled PET for synthesis of waterborne alkyd. The two themes, recycling of materials and waterborne coatings, involved in this paper are both the currently hot research topics. I think the paper is publishable.

Prior to publication, I suggest the authors to carry out the following revisions:

(1) The authors had better compare the properties of the recycled PET-based alkyd coatings with those of commercial waterborne alkyd coatings.

(2) I do not agree that the shear thinning of the waterborne alkyd resin was caused by the open of the network structure of the molecules. It should be noted that it was the polymer dispersion not the solution involved herein. The shear thinning may be more possibly caused by the destroying of polymer droplet network.

(3) There are many grammar and editing problem. The authors should improve their writing a lot. Some examples: The last sentence of the first paragraph in Introduction section was written in wrong grammar. Subscript and superscript were not cared about throughout the manuscript. The model of GPC instrument should be Waters 1515, not Waters 515. In Table 2, "alkyd rein" should be "alkyd resin". In section 3.2, line 12 of the first paragraph, "stem" should be "stemmed".

Review form: Reviewer 2

Is the manuscript scientifically sound in its present form?

No

Are the interpretations and conclusions justified by the results?

Yes

Is the language acceptable?

No

Do you have any ethical concerns with this paper?

No

Have you any concerns about statistical analyses in this paper?

No

Recommendation?

Major revision is needed (please make suggestions in comments)

Comments to the Author(s)

Please find here the comment for the manuscript RSOS-191447 entitled "Preparation and characterization of water-borne alkyd amino baking coatings based on waste PET". In this manuscript authors reported the synthesis waterborne alkyd amino baked coatings from the polyethylene terephthalate waste and evaluate their coating behavior by measuring it through standard tests. The Manuscript is good initiative to use the polyethylene terephthalate waste in coating industries to reduce petro based polymers and their use after the recycling process. The

manuscript is not properly written and have some serious language, presentation and novelty issues which may be justified before its publication in "Royal Society Open Science". So I recommend major revision this manuscript for its publication in Surface Review and Letters, I include my comment below:

- 1) The novelty of the manuscript is in doubt as I found similar papers on the same topic so I request the authors please highlight the difference or advancement of their work by giving the comparison table (for example <https://doi.org/10.1016/j.wasman.2008.02.018> ; <https://doi.org/10.1007/s00289-009-0166-4>, etc).
- 2) The manuscript requires a careful English language revision for grammar, spelling mistakes and its UK English or US English pattern author use and should be adhered to single pattern.
- 3) Font of the text should be uniform throughout the manuscript according to the guideline of the journal.
- 4) Proton NMR should be written as "1HNMR".
- 5) In NMR discussion highlight or underline the hydrogen under the discussion.
- 6) In NMR, What is HIPA and HTPA?

Regards

Review form: Reviewer 3

Is the manuscript scientifically sound in its present form?

No

Are the interpretations and conclusions justified by the results?

Yes

Is the language acceptable?

Yes

Do you have any ethical concerns with this paper?

No

Have you any concerns about statistical analyses in this paper?

No

Recommendation?

Major revision is needed (please make suggestions in comments)

Comments to the Author(s)

Comments to the Author(s)

This manuscript needs major revision. These revisions are presented below.

1. The number of literature is not enough for this topic. It is better if the number of the literature is increased. As it known, depolymerization of waste PET with glycolysis reaction have been carried out before many times in the literature. Depolymerization products of PET have been used for different fields such as preparation of unsaturated polyesters, polyurethanes, alkyd resins etc. or adsorbent for basic dyes from aqueous solutions. However, only a small number articles are given as references, in this manuscript. Therefore, it should be done literature survey in detail and related articles about reused of depolymerization product of PET should be given as reference.

In addition, there are also many of articles about this waterborne or water reducible alkyds in the literature. Therefore, it should be done literature survey in detail again and new articles should be added to the reference list. For your review, related articles that have been in the literature presented as list below.

Related articles that have been in the literature:

- * Acar I., Bal A., Güçlü G., 2013. "The Use of Intermediates obtained from Aminoglycolysis of Waste Poly(Ethylene Terephthalate) (PET) for the Synthesis of Water-Reducible Alkyd Resin", *Canadian J. Chem.*, 91, 357-363.
- * Acar I., Bal A., Güçlü G., 2013. "The Effect of Xylene as Aromatic Solvent to Aminoglycolysis of Post Consumer PET Bottles", *Polym. Eng. Sci.*, 53, 2429-2438.
- * Acar I., Bal A., Güçlü G., 2012. "Adsorption of Basic Dyes from Aqueous Solutions by Depolymerization Products of Post-Consumer PET Bottles", *Clean-Soil, Air, Water*, 40, 325-333.
- * Acar I., Kaşgöz A., Özgümüş S., Orbay M., 2006. "Modification of Waste Poly(Ethylene Terephthalate) (PET) by Using Poly(L-Lactic Acid) (PLA) and Hydrolytic Stability", *Polym.-Plast. Tech. Eng.*, 45, 351-359.
- * Akgün, N, Büyükyonga, ÖN, Acar, I, Güçlü, G, "Synthesis of Novel Acrylic Modified Water Reducible Alkyd Resin: Investigation of Acrylic Copolymer Ratio Effect on Film Properties and Thermal Behaviors." *Polym. Eng. Sci.*, 56:947-954 (2016).
- * Akbarinezhad, E, Ebrahimi, M, Kassirha, SM, Khorasani, "Synthesis and Evaluation of Water-Reducible Acrylic- Alkyd Resins with High Hydrolytic Stability." *Prog. Org. Coat.*, 65 217-221 (2009)
- * Aslan, S., Immirzi, B., Laurienzo, P., Malinconico, M., Martuscelli, E., Volpe, M.G. 1997, "Unsaturated Polyester Resins From Glycolysed Waste Polyethylene Terephthalate; Synthesis and Comparison of Properties and Performance with Virgin Resin", *J. Matter. Sci.*, 32, 2329-2336.
- * Athawale, VD, Nimbalkar, RV, "Waterborne Coatings Based on Renewable Oil Resources: an Overview." *J. Am. Oil Chem. Soc.*, 88 159-185 (2011)
- * Bal K., Ünlü K.C., Acar I., Güçlü G., 2017. "Epoxy Based Paints From Glycolysis Products of Post-Consumer PET Bottles: Synthesis, Wet Paint Properties and Film Properties", *J. Coat. Tech. Res.*, 14, 47-753.
- * Bulak E., Acar I., 2014. "The Use of Aminolysis, Aminoglycolysis and Simultaneous Aminolysis-Hydrolysis Products of Waste PET for Production of Paint Binder ", *Polym. Eng. Sci.*, 54, 2273-2281.
- * Büyükyonga, ÖN, Akgün, N, Acar, I, Güçlü, G, "Synthesis of four-component acrylic-modified water-reducible alkyd resin: investigation of dilution ratio effect on film properties and thermal behaviors." *J. Coat. Technol. Res.*, 14 (1) 117-128 (2017).
- * Ertaş K., Güçlü G. 2005. "Alkyd Resins Synthesized from Glycolysis Products of Waste PET", *Polym. Plast. Technol. Eng.*, 44, 783-794.
- * Farahat, M.S., Abdel-Azim, A.A., Abdel-Raowf, M.E. 2000. "Modified Unsaturated Polyester Resins Synthesized from Poly(Ethylene Terephthalate) Waste. 1. Synthesis And Curing Characteristics", *Macromol. Mater. Eng.*, 283, 1-6.
- * Güçlü, G., Kaşgöz, A., Özbudak, S., Özgümüş, S., Orbay, M. 1998. "Glycolysis of Poly (Ethylene Terephthalate) Wastes in Xylene", *J. Appl. Polym. Sci.*, 69, 2311-2319.
- * Güçlü, G., Orbay M. 2009. "Alkyd Resins Synthesized from Postconsumer PET Bottles", *Prog. Org. Coat.*, 65, 362-365, 2009.
- * Güçlü, G. 2010. "Alkyd Resins Based on Waste PET for Water-Reducible Coating Applications", *Polym. Bull.*, 64, 739-748.
- * Minari, RJ, Goikoetxea, M, Beristain, I, Paulis, M, Barandiaran, MJ, Asua, JM, "Post-polymerization of Waterborne Alkyd/Acrylics. Effect on Polymer Architecture and Particle Morphology." *Polymer*, 50 5892-5900 (2009)

- * Öztürk Y., Güçlü G. 2004. "Unsaturated Polyester Resins Obtained From Glycolysis Products of Waste PET", *Polym. Plast. Technol. Eng.*, 43, 1539-1552.
- * Paszun, D., Spychaj, T. 1997. "Chemical Recycling of Poly(Ethylene Terephthalate)", *Ind. Eng. Chem. Res.*, 36, 1373-1383.
- * Suh, D.J., Park, O.O., Yoon, K.H. 2000. "The Properties of Unsaturated Polyester based on the Glycolyzed Poly(Ethylene Terephthalate) with Various Glycol Compositions", *Polymer*, 41, 461-466.
- * Torlakoglu, A., Güçlü G. 2009. "Alkyd-Amino Resins Based on Waste PET for Coating Applications", *Waste Management*, 29, 350-354, 2009.
- * Vaidya, U.R., Nadkarni, V.M. 1987. "Unsaturated Polyester Resins From PET Waste: Kinetics of Polycondensation", *J. Appl. Polym. Sci.*, 34, 235-245.
- * Vaidya, U.R., Nadkarni, V.M. 1987. "Unsaturated Polyester Resins from Poly(Ethylene Terephthalate) Waste: Synthesis and Characterization, *Ind. Eng. Chem. Res.*, 26, 194-198.
- * Wang, C, Lin, G, Pae, JH, Jones, FN, Ye, H, Shen, W, "Novel Synthesis of Carboxy-Functional Soybean Acrylic- Alkyd Resins for Water-Reducible Coatings." *J. Coat. Technol.*, 72 55-61 (2000)
- * Yousefi, AA, Pishvaei, M, Yousefi, A, "Preparation of Water-Based Alkyd/ Acrylic Hybrid Resins." *Prog. Color Colourant Coat.*, 4 15-25 (2011)

2. The original aspect of this study and difference from other similar studies should be emphasized thoroughly in the introduction section. Similar studies should be summarized in a comparative table. Experimental results should be compared with similar studies which were given in the literature.

3. The abbreviations which were used in this study should be given in a separate table.

4. The experimental results should be sufficiently discussed and interpreted. The Conclusion section should be comprehensive in the light of the experiments done.

5. The English of all manuscript should be controlled and grammatical and stylistic errors should be corrected.

Decision letter (RSOS-191447.R0)

28-Oct-2019

Dear Dr Yong-bo:

Title: Preparation and characterization of water-borne alkyd-amino baking coatings based on waste PET

Manuscript ID: RSOS-191447

The editor assigned to your manuscript has now received comments from reviewers. We would like you to revise your paper in accordance with the referee and Subject Editor suggestions which can be found below (not including confidential reports to the Editor). Please note this decision does not guarantee eventual acceptance.

Please submit your revised paper before 20-Nov-2019. Please note that the revision deadline will expire at 00.00am on this date. If we do not hear from you within this time then it will be

assumed that the paper has been withdrawn. In exceptional circumstances, extensions may be possible if agreed with the Editorial Office in advance. We do not allow multiple rounds of revision so we urge you to make every effort to fully address all of the comments at this stage. If deemed necessary by the Editors, your manuscript will be sent back to one or more of the original reviewers for assessment. If the original reviewers are not available we may invite new reviewers.

RSC Associate Editor:
Comments to the Author:
(There are no comments.)

RSC Subject Editor:
Comments to the Author:
(There are no comments.)

Reviewers' Comments to Author:
Reviewer: 1

Comments to the Author(s)
In this paper, the authors partially used the glycolyzed PET product to synthesize waterborne alkyd resin. The resin was further fabricated as waterborne alkyd-amino baking coatings with

better mechanical properties. Although the glycolyzed PET was ever used to synthesize alkyd, it is seemed that this paper is the first time to employ the recycled PET for synthesis of waterborne alkyd. The two themes, recycling of materials and waterborne coatings, involved in this paper are both the currently hot research topics. I think the paper is publishable.

Prior to publication, I suggest the authors to carry out the following revisions:

- (1) The authors had better compare the properties of the recycled PET-based alkyd coatings with those of commercial waterborne alkyd coatings.
- (2) I do not agree that the shear thinning of the waterborne alkyd resin was caused by the open of the network structure of the molecules. It should be noted that it was the polymer dispersion not the solution involved herein. The shear thinning may be more possibly caused by the destroying of polymer droplet network.
- (3) There are many grammar and editing problem. The authors should improve their writing a lot. Some examples: The last sentence of the first paragraph in Introduction section was written in wrong grammar. Subscript and superscript were not cared about throughout the manuscript. The model of GPC instrument should be Waters 1515, not Waters 515. In Table 2, "alkyd rein" should be "alkyd resin". In section 3.2, line 12 of the first paragraph, "stem" should be "stemmed".

Reviewer: 2

Comments to the Author(s)

Please find here the comment for the manuscript RSOS-191447 entitled "Preparation and characterization of water-borne alkyd amino baking coatings based on waste PET". In this manuscript authors reported the synthesis waterborne alkyd amino baked coatings from the polyethylene terephthalate waste and evaluate their coating behavior by measuring it through standard tests. The Manuscript is good initiative to use the polyethylene terephthalate waste in coating industries to reduce petro based polymers and their use after the recycling process. The manuscript is not properly written and have some serious language, presentation and novelty issues which may be justified before its publication in "Royal Society Open Science". So I recommend major revision this manuscript for its publication in Surface Review and Letters, I include my comment below:

- 1) The novelty of the manuscript is in doubt as I found similar papers on the same topic so I request the authors please highlight the difference or advancement of their work by giving the comparison table (for example <https://doi.org/10.1016/j.wasman.2008.02.018> ; <https://doi.org/10.1007/s00289-009-0166-4>, etc).
- 2) The manuscript requires a careful English language revision for grammar, spelling mistakes and its UK English or US English pattern author use and should be adhered to single pattern.
- 3) Font of the text should be uniform throughout the manuscript according to the guideline of the journal.
- 4) Proton NMR should be written as "1HNMR".
- 5) In NMR discussion highlight or underline the hydrogen under the discussion.
- 6) In NMR, What is HIPA and HTPA?

Regards

Reviewer: 3

Comments to the Author(s)

This manuscript needs major revision. These revisions are presented below.

1. The number of literature is not enough for this topic. It is better if the number of the literature is increased. As it known, depolymerization of waste PET with glycolysis reaction have been

carried out before many times in the literature. Depolymerization products of PET have been used for different fields such as preparation of unsaturated polyesters, polyurethanes, alkyd resins etc. or adsorbent for basic dyes from aqueous solutions. However, only a small number articles are given as references, in this manuscript. Therefore, it should be done literature survey in detail and related articles about reused of depolymerization product of PET should be given as reference.

In addition, there are also many of articles about this waterborne or water reducible alkyds in the literature. Therefore, it should be done literature survey in detail again and new articles should be added to the reference list. For your review, related articles that have been in the literature presented as list below.

Related articles that have been in the literature:

- * Acar I., Bal A., Güçlü G., 2013. "The Use of Intermediates obtained from Aminoglycolysis of Waste Poly(Ethylene Terephthalate) (PET) for the Synthesis of Water-Reducible Alkyd Resin", Canadian J. Chem., 91, 357-363.
- * Acar I., Bal A., Güçlü G., 2013. "The Effect of Xylene as Aromatic Solvent to Aminoglycolysis of Post Consumer PET Bottles", Polym. Eng. Sci., 53, 2429-2438.
- * Acar I., Bal A., Güçlü G., 2012. "Adsorption of Basic Dyes from Aqueous Solutions by Depolymerization Products of Post-Consumer PET Bottles", Clean-Soil, Air, Water, 40, 325-333.
- * Acar I., Kaşgöz A., Özgümüş S., Orbay M., 2006. "Modification of Waste Poly(Ethylene Terephthalate) (PET) by Using Poly(L-Lactic Acid) (PLA) and Hydrolytic Stability", Polym.-Plast. Tech. Eng., 45, 351-359.
- * Akgün, N, Büyükyonga, ÖN, Acar, I, Güçlü, G, "Synthesis of Novel Acrylic Modified Water Reducible Alkyd Resin: Investigation of Acrylic Copolymer Ratio Effect on Film Properties and Thermal Behaviors." Polym. Eng. Sci., 56:947-954 (2016).
- * Akbarinezhad, E, Ebrahimi, M, Kassiriha, SM, Khorasani, "Synthesis and Evaluation of Water-Reducible Acrylic- Alkyd Resins with High Hydrolytic Stability." Prog. Org. Coat., 65 217-221 (2009)
- * Aslan, S., Immirzi, B., Laurienzo, P., Malinconico, M., Martuscelli, E., Volpe, M.G. 1997, "Unsaturated Polyester Resins From Glycolysed Waste Polyethylene Terephthalate; Synthesis and Comparison of Properties and Performance with Virgin Resin", J. Matter. Sci., 32, 2329-2336.
- * Athawale, VD, Nimbalkar, RV, "Waterborne Coatings Based on Renewable Oil Resources: an Overview." J. Am. Oil Chem. Soc., 88 159-185 (2011)
- * Bal K., Ünlü K.C., Acar I., Güçlü G., 2017. "Epoxy Based Paints From Glycolysis Products of Post-Consumer PET Bottles: Synthesis, Wet Paint Properties and Film Properties", J. Coat. Tech. Res., 14, 47-753.
- * Bulak E., Acar I., 2014. "The Use of Aminolysis, Aminoglycolysis and Simultaneous Aminolysis-Hydrolysis Products of Waste PET for Production of Paint Binder", Polym. Eng. Sci., 54, 2273-2281.
- * Büyükyonga, ÖN, Akgün, N, Acar, I, Güçlü, G, "Synthesis of four-component acrylic-modified water-reducible alkyd resin: investigation of dilution ratio effect on film properties and thermal behaviors." J. Coat. Technol. Res., 14 (1) 117-128 (2017).
- * Ertaş K., Güçlü G. 2005. "Alkyd Resins Synthesized from Glycolysis Products of Waste PET", Polym. Plast. Technol. Eng., 44, 783-794.
- * Farahat, M.S., Abdel-Azim, A.A., Abdel-Raouf, M.E. 2000. "Modified Unsaturated Polyester Resins Synthesized from Poly(Ethylene Terephthalate) Waste. 1. Synthesis And Curing Characteristics", Macromol. Mater. Eng., 283, 1-6.
- * Güçlü, G., Kaşgöz, A., Özbudak, S., Özgümüş, S., Orbay, M. 1998. "Glycolysis of Poly (Ethylene Terephthalate) Wastes in Xylene", J. Appl. Polym. Sci., 69, 2311-2319.
- * Güçlü, G., Orbay M. 2009. "Alkyd Resins Synthesized from Postconsumer PET Bottles", Prog. Org. Coat., 65, 362-365, 2009.

- * Güçlü, G. 2010. "Alkyd Resins Based on Waste PET for Water-Reducible Coating Applications", *Polym. Bull.*, 64, 739-748.
- * Minari, RJ, Goikoetxea, M, Beristain, I, Paulis, M, Barandiaran, MJ, Asua, JM, "Post-polymerization of Waterborne Alkyd/Acrylics. Effect on Polymer Architecture and Particle Morphology." *Polymer*, 50 5892-5900 (2009)
- * Öztürk Y., Güçlü G. 2004. "Unsaturated Polyester Resins Obtained From Glycolysis Products of Waste PET", *Polym. Plast. Technol. Eng.*, 43, 1539-1552.
- * Paszun, D., Spychaj, T. 1997. "Chemical Recycling of Poly(Ethylene Terephthalate)", *Ind. Eng. Chem. Res.*, 36, 1373-1383.
- * Suh, D.J., Park, O.O., Yoon, K.H. 2000. "The Properties of Unsaturated Polyester based on the Glycolized Poly(Ethylene Terephthalate) with Various Glycol Compositions", *Polymer*, 41, 461-466.
- * Torlakoglu, A., Güçlü G. 2009. "Alkyd-Amino Resins Based on Waste PET for Coating Applications", *Waste Management*, 29, 350-354, 2009.
- * Vaidya, U.R., Nadkarni, V.M. 1987. "Unsaturated Polyester Resins From PET Waste: Kinetics of Polycondensation", *J. Appl. Polym. Sci.*, 34, 235-245.
- * Vaidya, U.R., Nadkarni, V.M. 1987. "Unsaturated Polyester Resins from Poly(Ethylene Terephthalate) Waste: Synthesis and Characterization, *Ind. Eng. Chem. Res.*, 26, 194-198.
- * Wang, C, Lin, G, Pae, JH, Jones, FN, Ye, H, Shen, W, "Novel Synthesis of Carboxy-Functional Soybean Acrylic- Alkyd Resins for Water-Reducible Coatings." *J. Coat. Technol.*, 72 55-61 (2000)
- * Yousefi, AA, Pishvaei, M, Yousefi, A, "Preparation of Water-Based Alkyd/Acrylic Hybrid Resins." *Prog. Color Colourant Coat.*, 4 15-25 (2011)

2. The original aspect of this study and difference from other similar studies should be emphasized thoroughly in the introduction section. Similar studies should be summarized in a comparative table. Experimental results should be compared with similar studies which were given in the literature.

3. The abbreviations which were used in this study should be given in a separate table.

4. The experimental results should be sufficiently discussed and interpreted. The Conclusion section should be comprehensive in the light of the experiments done.

5. The English of all manuscript should be controlled and grammatical and stylistic errors should be corrected.

Author's Response to Decision Letter for (RSOS-191447.R0)

See Appendix A.

Decision letter (RSOS-191447.R1)

21-Nov-2019

Dear Dr Yong-bo:

Title: Preparation and characterization of water-borne alkyd-amino baking coatings based on waste PET

Manuscript ID: RSOS-191447.R1

It is a pleasure to accept your manuscript in its current form for publication in Royal Society Open Science. The chemistry content of Royal Society Open Science is published in collaboration with the Royal Society of Chemistry.

RSC Associate Editor
Comments to the Author:
(There are no comments.)

Reviewer(s)' Comments to Author:

Appendix A

Dear Editors

Thank you for your letter and for the reviewers' comments concerning our manuscript (Manuscript ID: RSOS-191447) entitled "Preparation and characterization of waterborne alkyd-amino baking coatings based on waste PET". Those comments are all valuable and very helpful for revising and improving our paper, as well as the important guiding significance to our researches. We have studied comments carefully and have made correction which we hope meet with approval. Revised portion are marked in red in the revised manuscript.

We greatly appreciate your continued interest in our manuscript.

Best wishes,

Sincerely yours,

Yong-Bo Ding

The main corrections in the paper and the responds to the reviewer's comments are as following:

Responds to the reviewer's comments:

Reviewer #1:

Comments to the Author(s)

In this paper, the authors partially used the glycolyzed PET product to synthesize waterborne alkyd resin. The resin was further fabricated as waterborne alkyd-amino baking coatings with better mechanical properties. Although the glycolyzed PET was ever used to synthesize alkyd, it is seemed that this paper is the first time to employ the recycled PET for synthesis of waterborne alkyd. The two themes, recycling of materials and waterborne coatings, involved in this paper are both the currently hot research topics. I think the paper is publishable.

Prior to publication, I suggest the authors to carry out the following revisions:

(1) The authors had better compare the properties of the recycled PET-based alkyd coatings with those of commercial waterborne alkyd coatings.

Response: Yes, the properties of the recycled PET-based alkyd coatings had been compared with those of commercial waterborne alkyd coatings, please see table 3 and table 4 in the revised manuscript.

(2) I do not agree that the shear thinning of the waterborne alkyd resin was caused by the open of the network structure of the molecules. It should be noted that it was the polymer dispersion not the solution involved herein. The shear thinning may be more possibly caused by the destroying of polymer droplet network.

Response: Yes, the shear thinning may be more possibly caused by the destroying of polymer droplet network, the reason for the shear thinning of the waterborne alkyd resin had been revised, please see the revised manuscript.

(3) There are many grammar and editing problem. The authors should improve their writing a lot. Some examples: The last sentence of the first paragraph in Introduction section was written in wrong grammar. Subscript and superscript were not cared about throughout the manuscript. The model of GPC instrument should be Waters 1515, not Waters 515. In Table 2, “alkyd rein” should be “alkyd resin”. In section 3.2, line 12 of the first paragraph, “stem” should be “stemmed”.

Response: We are very sorry for our incorrect grammar and editing problem. These problems have been corrected in the revised manuscript.

Reviewer: 2

Comments to the Author(s)

Please find here the comment for the manuscript RSOS-191447 entitled “Preparation and characterization of water-borne alkyd amino baking coatings based on waste PET”. In this manuscript authors reported the synthesis waterborne alkyd amino baked coatings from the polyethylene terephthalate waste and evaluate their coating behavior

by measuring it through standard tests. The Manuscript is good initiative to use the polyethylene terephthalate waste in coating industries to reduce petro based polymers and their use after the recycling process. The manuscript is not properly written and have some serious language, presentation and novelty issues which may be justified before its publication in “Royal Society Open Science”. So I recommend major revision this manuscript for its publication in Surface Review and Letters, I include my comment below:

1) The novelty of the manuscript is in doubt as I found similar papers on the same topic so I request the authors please highlight the difference or advancement of their work by giving the comparison table (for example <https://doi.org/10.1016/j.wasman.2008.02.018>; <https://doi.org/10.1007/s00289-009-0166-4>, etc).

Response: We are very sorry for we didn't explain our experimental steps clearly, so in the part of 2.4(preparation of waterborne alkyd–amino baking coating), the dilution process for waterborne alkyd resin is listed, please see the red part in the revised version. In addition, the comparison table was given, please see table 1.

Table 1 Classification, VOC content and surfactant of alkyd resin

Reference	Classification	VOC content (weight)	Surfactant
doi.org/10.1016/j.wasman.2008.02.018	Solvent-borne resin	40% xylene	Without
doi.org/10.1007/s00289-009-0166-4	Water-borne resin	30% isopropyl alcohol	Tween 60
In this manuscript	Water-borne resin	20% dipropylene glycol butyl ether	Without

According to table 1, the difference or advancement of our work are obvious.

2) The manuscript requires a careful English language revision for grammar, spelling mistakes and its UK English or US English pattern author use and should be adhered to single pattern.

Response: We are very sorry for our incorrect grammar and editing problem. These problems have been corrected in the revised manuscript.

3) Font of the text should be uniform throughout the manuscript according to the guideline of the journal.

Response: We are very sorry for our incorrect uniform. These problems have been corrected in the revised manuscript.

4) Proton NMR should be written as “¹HNMR”.

Response: Yes, proton NMR had been written as “¹HNMR”.

5) In NMR discussion highlight or underline the hydrogen under the discussion.

Response: Yes, in NMR discussion, the hydrogen under the discussion had been underlined.

6) In NMR, What is HIPA and HTPA?

Response: We are very sorry for our wrong ways, HIPA and HTPA had been revised.

Reviewer: 3

Comments to the Author(s)

This manuscript needs major revision. These revisions are presented below.

1. The number of literature is not enough for this topic. It is better if the number of the literature is increased. As it known, depolymerization of waste PET with glycolysis reaction have been carried out before many times in the literature. Depolymerization products of PET have been used for different fields such as preparation of unsaturated polyesters, polyurethanes, alkyd resins etc. or adsorbent for basic dyes from aqueous

solutions. However, only a small number of articles are given as references, in this manuscript. Therefore, it should be done literature survey in detail and related articles about reused of depolymerization product of PET should be given as reference.

In addition, there are also many of articles about this waterborne or water reducible alkyds in the literature. Therefore, it should be done literature survey in detail again and new articles should be added to the reference list. For your review, related articles that have been in the literature presented as list below.

Related articles that have been in the literature:

* Acar I., Bal A., Güçlü G., 2013. "The Use of Intermediates obtained from Aminoglycolysis of Waste Poly(Ethylene Terephthalate) (PET) for the Synthesis of Water-Reducible Alkyd Resin", *Canadian J. Chem.*, 91, 357-363.

* Acar I., Bal A., Güçlü G., 2013. "The Effect of Xylene as Aromatic Solvent to Aminoglycolysis of Post Consumer PET Bottles", *Polym. Eng. Sci.*, 53, 2429-2438.

* Acar I., Bal A., Güçlü G., 2012. "Adsorption of Basic Dyes from Aqueous Solutions by Depolymerization Products of Post-Consumer PET Bottles", *Clean-Soil, Air, Water*, 40, 325-333.

* Acar I., Kaşgöz A., Özgümüş S. , Orbay M., 2006. "Modification of Waste Poly(Ethylene Terephthalate) (PET) by Using Poly(L-Lactic Acid) (PLA) and Hydrolytic Stability", *Polym.-Plast. Tech. Eng.*, 45, 351-359.

* Akgün, N, Büyükkyonga, ÖN, Acar, I, Güçlü, G, "Synthesis of Novel Acrylic Modified Water Reducible Alkyd Resin: Investigation of Acrylic Copolymer Ratio Effect on Film Properties and Thermal Behaviors." *Polym. Eng. Sci.*, 56:947–954 (2016).

* Akbarinezhad, E, Ebrahimi, M, Kassiriha, SM, Khorasani, "Synthesis and Evaluation of Water-Reducible Acrylic- Alkyd Resins with High Hydrolytic Stability." *Prog. Org. Coat.*, 65 217–221 (2009)

* Aslan, S., Immirzi, B., Laurienzo, P., Malinconico, M., Martuscelli, E., Volpe, M.G.

1997, "Unsaturated Polyester Resins From Glycolysed Waste Polyethylene Terephthalate; Synthesis and Comparison of Properties and Performance with Virgin Resin", *J. Matter. Sci.*, 32, 2329-2336.

* Athawale, VD, Nimbalkar, RV, "Waterborne Coatings Based on Renewable Oil Resources: an Overview." *J. Am. Oil Chem. Soc.*, 88 159–185 (2011)

* Bal K., Ünlü K.C. , Acar I., Güçlü G., 2017. "Epoxy Based Paints From Glycolysis Products of Post-Consumer PET Bottles: Synthesis, Wet Paint Properties and Film Properties", *J. Coat. Tech. Res.*, 14, 47-753.

* Bulak E., Acar I., 2014. "The Use of Aminolysis, Aminoglycolysis and Simultaneous Aminolysis-Hydrolysis Products of Waste PET for Production of Paint Binder ", *Polym. Eng. Sci.*, 54, 2273-2281.

* Büyükkyonga, ÖN, Akgün, N, Acar, I, Güçlü, G, "Synthesis of four-component acrylic-modified water-reducible alkyd resin: investigation of dilution ratio effect on film properties and thermal behaviors." *J. Coat. Technol. Res.*, 14 (1) 117–128 (2017).

* Ertaş K., Güçlü G. 2005. "Alkyd Resins Synthesized from Glycolysis Products of Waste PET", *Polym. Plast. Technol. Eng.*, 44, 783-794.

* Farahat, M.S., Abdel-Azim, A.A., Abdel-Raouf, M.E. 2000. "Modified Unsaturated Polyester Resins Synthesized from Poly(Ethylene Terephthalate) Waste. 1. Synthesis And Curing Characteristics", *Macromol. Mater. Eng.*, 283, 1-6.

* Güçlü, G., Kaşgöz, A., Özbudak, S., Özgümüş, S., Orbay, M. 1998. "Glycolysis of Poly (Ethylene Terephthalate) Wastes in Xylene", *J. Appl. Polym. Sci.*, 69, 2311-2319.

* Güçlü, G., Orbay M. 2009. "Alkyd Resins Synthesized from Postconsumer PET Bottles", *Prog. Org. Coat.*, 65, 362-365, 2009.

* Güçlü, G. 2010. "Alkyd Resins Based on Waste PET for Water-Reducible Coating Applications", *Polym. Bull.*, 64, 739–748.

* Minari, RJ, Goikoetxea, M, Beristain, I, Paulis, M, Barandiaran, MJ, Asua, JM, "Post-polymerization of Waterborne Alkyd/Acrylics. Effect on Polymer Architecture and Particle Morphology." *Polymer*, 50 5892–5900 (2009)

* Öztürk Y., Güçlü G. 2004. "Unsaturated Polyester Resins Obtained From Glycolysis Products of Waste PET", *Polym. Plast. Technol. Eng.*, 43, 1539-1552.

- * Paszun, D., Spychaj, T. 1997. "Chemical Recycling of Poly(Ethylene Terephthalate)", Ind. Eng. Chem. Res., 36, 1373-1383.
- * Suh, D.J., Park, O.O., Yoon, K.H. 2000. "The Properties of Unsaturated Polyester based on the Glycolyzed Poly(Ethylene Terephthalate) with Various Glycol Compositions", Polymer, 41, 461-466.
- * Torlakoğlu, A., Güçlü G. 2009. "Alkyd-Amino Resins Based on Waste PET for Coating Applications", Waste Management, 29, 350-354, 2009.
- * Vaidya, U.R., Nadkarni, V.M. 1987. "Unsaturated Polyester Resins From PET Waste: Kinetics of Polycondensation", J. Appl. Polym. Sci., 34, 235-245.
- * Vaidya, U.R., Nadkarni, V.M. 1987. "Unsaturated Polyester Resins from Poly(Ethylene Terephthalate) Waste: Synthesis and Characterization, Ind. Eng. Chem. Res., 26, 194-198.
- * Wang, C, Lin, G, Pae, JH, Jones, FN, Ye, H, Shen, W, "Novel Synthesis of Carboxy-Functional Soybean Acrylic- Alkyd Resins for Water-Reducible Coatings." J. Coat. Technol., 72 55–61 (2000)
- * Yousefi, AA, Pishvaei, M, Yousefi, A, "Preparation of Water-Based Alkyd/Acrylic Hybrid Resins." Prog. Color Colourant Coat., 4 15–25 (2011)

Response: Yes, the number of the literature is increased in revised manuscript.

2. The original aspect of this study and difference from other similar studies should be emphasized thoroughly in the introduction section. Similar studies should be summarized in a comparative table. Experimental results should be compared with similar studies which were given in the literature.

Response: Yes, similar studies had been emphasized thoroughly in the introduction section, and experimental results of similar studies had been summarized in table 3 and table 4.

3. The abbreviations which were used in this study should be given in a separate table.

Response: Yes, the abbreviations had been given in a separate table, please see part of abbreviations and corresponding full names in revised manuscript.

Abbreviations and corresponding full names

Abbreviations	Corresponding full names
PET	polyethylene terephthalate
VOC	volatile organic component
TOFA	Tall oil fatty acids
NPG	Neopentyl glycol
IPA	isophthalic acid
TPA	terephthalic acid
BA	benzoic acid
TMA	trimellitic anhydride
PE	Pentaerythritol
DMEA	N, N-Dimethylethanolamine
DPNB	dipropylene glycol butyl ether
BCS	butyl cellosolve
AN	acid number

4. The experimental results should be sufficiently discussed and interpreted. The Conclusion section should be comprehensive in the light of the experiments done.

Response: Yes, the experimental results had been sufficiently discussed and interpreted.

The conclusion section is comprehensive in the light of the experiments done.

5. The English of all manuscript should be controlled and grammatical and stylistic errors should be corrected.

Response: Yes, the English of all manuscript had been controlled and grammatical and stylistic errors had been corrected.